# The Impact of Environmental Regulation on the Green Overall Factor Productivity of Forestry in the Yangtze River Economic Belt

Deli Li, Yang Li, Gang Tian * and Richard K. Mendako

College of Economics and Management, Northeast Forestry University, Harbin 150040, China; lideli1964@nefu.edu.cn (D.L.); liyang502921@163.com (Y.L.); richardmendako@gmail.com (R.K.M.)
* Correspondence: tiangang0451@nefu.edu.cn; Tel.: +86-1894-6059-819

**Abstract:** Environmental regulatory instruments are key to achieving synergy between high-quality economic development and ecological civilization construction. This paper measures the green overall factor productivity of the forestry in the Yangtze River Economic Belt by using the super-efficient nonexpected output SBM-ML index model. Additionally, it investigates the environmental regulation's impact on forestry's overall green factor productivity by using the conventional panel regression and panel smooth transformation model. The model was based on the relevant data obtained from eleven provinces along the Yangtze River Economic Belt in China from 2006 to 2021. This study concludes that command-and-control regulation of the environment and public engagement with environmental regulation can significantly promote the forestry green overall factor productivity in the Yangtze River Economic Belt, and the environmental regulation's effects on the forestry green overall efficiency in the economic region of the Yangtze River are regionally and temporally heterogeneous. The command-and-control environmental regulation also needs to exceed a certain level of regulatory coercion to promote the forestry green overall factor productivity positively. The effects of market-incentive environmental regulation were more pronounced with the increase in the regulatory intensity. When the regulation intensity surpasses the threshold, the public participation form of environmental regulation has a depressing impact on the forestry green overall factor output. To promote the development of the forestry industry in the Yangtze River Economic Zone, it is therefore necessary to strengthen the coordination of different environmental regulations, implement measures in each region, build a market-oriented green innovation system, and promote the structuralization and upgrading of the forestry industry.

**Keywords:** Yangtze River Economic Belt; environmental regulation; green productivity of overall factors





## 1. Introduction

### 1.1. Background of Topic Selection

The Chinese Government has put forward the concept of "lucid waters and lush mountains are invaluable assets" and the vision of building a beautiful China, insisting on the harmonious coexistence of human beings and nature. Forestry is of great significance to the construction of ecological civilization, and its role in promoting the green and sustainable development of the region cannot be ignored. The overall factor productivity is an important indicator for assessing the sustainable development and operational quality of the forestry economy, which can accurately reflect the development status of the forestry industry [1]. China's Yangtze River Economic Belt is an important ecological security barrier zone that spans the east, central, and west sectors of China. It covers eleven provinces and municipalities, accounting for about one fifth of the country's land area, with unique geographic advantages and great development potential. The Yangtze River Economic Belt is one of the three major strategies for China's regional development in the new era

considering that it has been highly valued by the Chinese government as the most densely populated region in Chinese society, with the largest scale of industrial development, the most developed level of economic management, and the most complete system of urban culture. In 2014, the State Council issued the "Guiding Opinions on Promoting the Development of the Yangtze River Economic Belt by Relying on the Golden Waterway", which explicitly proposes to make the Yangtze River Economic Belt become a demonstration area of ecological civilization construction with high-quality economic benefits and good ecology, and to realize the economy's development towards green, ecological, circular, and low-carbon development. However, the current overuse and disorderly development of natural resources in China's Yangtze River Economic Belt has degraded the ecosystem service function in some areas so severely that the original forest vegetation has been destroyed and the forest ecosystem degraded [2]. For this reason, there is a need for the Chinese government to enhance the coordination between promoting high-quality economic development and ensuring high-level protection of the ecological environment. The Yangtze River Economic Belt area (Figure 1) is depicted on the map below.

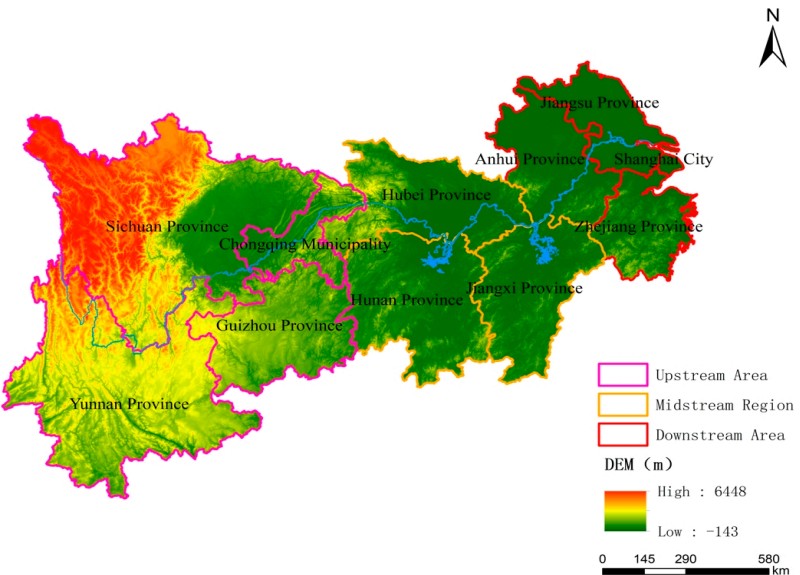

**Figure 1.** Regional geographic map of the Yangtze River Economic Belt.

Environmental regulation serves as a constraining force to deepen the environmental governance's effect and promote the construction of ecological civilization by targeting social individuals or organizations, which plays an important role in enhancing the synergy between environmental friendliness and economic prosperity [3]. China's Yangtze River Economic Zone should rely on environmental regulation tools to realize the ambitious goal of combining forestry economic prosperity and ecological environmental protection. Based on this, exploring the differences between effective environmental regulation tools and appropriate environmental regulation levels on forestry's green overall factor productivity can guide the direction of environmental regulation adjustments and innovations. Additionally, it can provide a reference for China's forestry industry to achieve green development and accelerate ecological civilization construction [4].

### 1.2. Research Significance

Environmental forest management is fundamental to ecological balance, the realization of peak carbon and carbon neutrality goals, and sustainable economic and social development. It has received close attention from governments all over the world. However, achieving rapid development of the forestry economy while ensuring that the consumption of resources and ecological and environmental pollution are within a reasonable range

is an issue that needs urgent attention. In this context, it is impossible to overlook how environmental regulation affects forestry's overall factor productivity [5].

From a theoretical perspective, the impact of environmental regulation on forestry eco-efficiency is multidimensional, and its impact may show different characteristics with the intensity of environmental regulation at different intervals. Based on this, the present paper focuses on exploring the linear and nonlinear impacts of different types of environmental regulation on forestry's overall green factor productivity. To address the conflict between the forestry economic value and ecological value, as well as the spatiotemporal variability in environmental regulations' influence on forestry green factor productivity, environmental pollution, and energy consumption, there is a need to accelerate the transformation of the forestry economic growth mode from traditional factors to advanced production factors and promote the coordinated and high-quality development of the regional green forestry industry.

Meanwhile, forestry is an important material-production sector and social public utility. Forestry development is a crucial component for sustainable economic and social growth. Environmental regulation is binding on social individuals or organizations. It can deepen the effect of environmental governance, promote the construction of ecological civilization, and help lead the coordinated development of economic prosperity and environmental friendliness. The Yangtze River Economic Belt can achieve its core objective of economic success and ecological environmental protection by applying environmental control. Based on the preliminary understanding of the sustainable development level of forestry in the Yangtze River Economic Belt, studying the impact of different types of effects of environmental regulations on forestry output in the Yangtze River Economic Belt in terms of the green overall factor can help provide a reliable scientific basis for the formulation of relevant policies and measures by government departments [6]. Analyzing the spatial and temporal heterogeneity of the impact of environmental regulations on the green overall factor productivity of forestry in the Yangtze River Economic Belt is key to putting forward scientific, targeted, and operational countermeasures for different types of environmental regulations. This could also aid in promoting the enhancement of the green overall factor productivity of forestry in the Yangtze River Economic Belt.

*1.3. Purpose of the Study*

This paper explores the spatial and temporal heterogeneity of the effects of environmental regulations on the overall green factor productivity of forestry in the Yangtze River Economic Belt, a significant green barrier. This study conducts an objective analysis of the effects of environmental regulations on the green general factor productivity in forestry and their potential mechanisms. Meanwhile, some policy recommendations are provided to promote the Yangtze River Economic Belt's economic prosperity and green sustainable development [7]. Specifically, it can be summarized as follows:

The forestry input–output index system was established based on the full consideration of the green, ecological, and economic benefits. The nonradial Slack-Based Measure–Malmquist Luenberger (SBM-ML) model was used to calculate the green overall factor productivity of forestry in the Yangtze River Economic Belt to accurately understand the current level of the sustainable development of the forestry industry in the Yangtze River Economic Belt.

An analysis of the effects of various environmental regulations on the overall green productivity of forestry in the Economic Belt of the Yangtze River was conducted. A panel data model was set up to look into the linear impact mechanism of various environmental regulations on the green overall factor productivity of forestry.

According to previous studies, for the temporal and spatial heterogeneity of the impact of environmental regulation on the green overall factor productivity of forestry in the Yangtze River Economic Belt, respective policies can be classified according to the characteristics of the upper, middle, and lower reaches of the Yangtze River. Different types of environmental regulation tools can be classified according to local conditions and the

resource and environmental endowments of different regions. The temporal evolution characteristics of the impact of environmental regulation on the forestry green overall factor productivity in the Yangtze River Economic Belt are then summarized.

We utilize the panel smooth transition model to further investigate the nonlinear characteristics of the influence of environmental regulations on the overall productivity of the green factors of forestry in the Yangtze River Economic Zone and to summarize the specific impact mechanisms of various environmental regulations on this productivity.

## 2. Literature Review

The effect of environmental regulations on the overall factor production has been extensively studied. For instance, Andrei et al., according to the environmental taxation in Romania, found that an environmental taxation policy can improve the environmental quality and increase the gross national product (GNP) to promote the local economy's green development [8]. Jorgenson and Wilcoxen and Chintrakam found that once a country imposes environmental regulations, enterprises will be forced to pay a certain amount of money to reduce or control environmental pollution to respond to the call, reducing corporate profits [9]. Bruunermerier and Levinson used panel data from 25 countries to investigate how the intensity of environmental regulations affected the industrial green overall factor productivity at the national level. The study revealed that environmental regulations inhibit the industrial green overall factor productivity [10]. Based on panel data on industrial firms in the US, Gray found that environmental regulation reduces the overall factor productivity of the firms [11]. Lanoie et al. used data from 17 manufacturing industries in Canada and found that the effect of environmental regulation on productivity was negative in the same period [12]. A similar study on the Yellow River Basin's economic environmental regulations' effects on the green general factor productivity revealed a U-type connection between environmental regulations and the green overall factor productivity [13].

For the study of the green overall factor productivity in forestry, J. Peter Clinchu introduced the production efficiency function and valuation method to study the general factor productivity in forestry, according to which the social efficiency of the forestry program bill of the government of Ireland was evaluated ex ante [14]. Gouranga G. Das et al. and others divided the US into four regions. Through the multiregional computational general equilibrium model, they concluded that technological change could improve the forestry output and increase welfare [15]. Other studies have examined timber forestry in Canadian regions, and the analysis of input–output efficiencies concluded that increasing the overall factor productivity in forestry can improve industrial competitiveness and increase the prices [16]. Luis DiazBalteiro et al. combined logistic regression with DEA and the analysis of the relationship between the technologies used in the Spanish timber industry. They concluded that increasing their drilling of internal technology can improve competitiveness [17]. Sporcic M et al. used DEA modeling to evaluate the input–output efficiency of the Croatian forest system and found that regional differences in the forest tenure and the degree of the intensive management of forest subgroups affect the forest ecoefficiency [18].

Meanwhile, environmental regulations' impact on the forestry industry's development is still controversial, and relatively little research has been conducted. Pan Dan found that market-incentive-based environmental regulation has a more significant and long-term effect on the afforestation area than command-and-control environmental regulations by using the double-difference method [19]. Xu Yuxiang et al. found that effective environmental regulations can optimize the ecological environment, thus affecting forestry's overall green factor productivity [20]. Jiang Wei et al. found that when the intensity of environmental regulation is low, it promotes the ecoefficiency of the forestry industry. At the same time, too high a power of environmental regulation will have the opposite effect [21]. Zheng, Y found that environmental regulation can optimize the forestry ecological environment by regulating FDI, which impacts how the forestry business develops [22].

In general, research on the overall factor productivity in forestry has been widely conducted by domestic and international scholars. However, there are still limited studies on the green overall factor productivity in forestry [23–25]. On the other hand, the relationship between environmental regulation and green development has consequently become a popular research issue for academics both domestically and internationally. This article calculates the green overall factor productivity of forestry in the Yangtze River Economic Belt in China by using the unexpected output super efficiency SBM-ML index model based on province data along the Yangtze River Economic Belt in China from 2006 to 2021. To investigate the spatiotemporal heterogeneity, linear and nonlinear aspects of the impact of environmental regulations on the green overall factor productivity of forestry in the Yangtze River Economic Belt in China have been used. The panel data model and panel smooth transformation model were employed to serve as a benchmark for ecological environment conservation as the forestry sector develops.

## 3. Materials and Methods

### 3.1. Research Hypothesis

Command-and-control environmental regulation is relatively mandatory and has a more significant role in promoting the development of the forestry industry in China. The government of China introduced appropriate laws and regulations, which require enterprises to meet more stringent emission standards. Under the pressure of such pollution control, enterprises are forced to transform and upgrade, increasing the cost burden. On the other hand, a part of the backward production capacity corresponds to the elimination, thus realizing the reduction in pollution. Nonetheless, the time and productivity can be increased to alleviate or even offset the costs brought about by environmental regulation.

Market-incentive-type environmental regulation applies economic incentives to increase the production costs of enterprises. Enterprises are forced to control pollution and technological innovation. In the long run, enterprises that cannot meet the emission standards will be restricted from entering the industry, while enterprises that do not meet the green production regulations will be eliminated, thus realizing an increase in the industry's overall green factor productivity. Public participatory environmental regulation, which is relatively noncompulsory and informal, is mainly realized through petitions and media reports [26].

Enterprises can increase public support by ensuring that they cater to public demands, reduce pollutant emissions, and realize the increase in sales and market share, thus improving the overall green factor productivity of the industry. However, the public's environmental protection social value system in China needs to be further improved, and the public's power to participate in environmental governance is not effectively guaranteed [27]. Based on this, this paper proposes the following hypothesis:

**H1:** *There are differences in the direct effect of diverse types of environmental regulation on the green overall factor productivity of forestry in China's Yangtze River Economic Belt.*

According to existing research, environmental regulation in the forestry industry's production costs can be distinguished as pollution control and technological innovation costs. Enterprises will be based on the actual situation of these two aspects of the dynamic adjustment, which will inhibit or promote the forestry green overall factor productivity.

Environmental regulation has a restraining effect on the overall factor productivity. On the one hand, when the government imposes environmental regulations, enterprises must spend money on transformation and upgrading, which generates corresponding environmental governance costs. These environmental governance costs will often crowd out the normal productive investments of enterprises, limiting the R&D investments of enterprises and failing to provide strong technological support for optimizing the green overall factor productivity of forestry. Overall, the cost loss inhibits the improvement in the forestry green overall factor productivity [28].

Environmental regulation is expected to have a promoting effect on the overall factor productivity. With the increasing intensity of environmental regulation, enterprises will spontaneously develop clean technologies and realize green production from a long-term perspective, thus enhancing the resource utilization efficiency. The resulting economic benefits could exceed the cost of environmental regulation. With the promotion of the environmental protection concept, consumer preference for green products will continue to increase, making enterprise products more competitive. Therefore, environmental regulation can protect the environment while simultaneously enhancing economic efficiency [29]. Based on this, this paper proposes the following hypothesis:

**H2:** *Environmental regulation has a nonlinear effect on the forestry industry's overall green factor productivity in China's Yangtze River Economic Belt.*

### 3.2. Description of Research Methodology and Variables

#### 3.2.1. SBM-ML Index Model for Unexpected Output Super Efficiency

In this paper, the super-efficient SBM model with a nonexpected output can effectively distinguish the efficiency differences between production decision-making units based on improving the traditional data envelopment analysis method, which cannot introduce slack variables into the objective function. For this reason, this paper adopts the super-efficient SBM model with a nonexpected output and the ML index model to measure forestry's overall green factor productivity in China's Yangtze River Economic Belt. Each forestry industry is considered as a production-decision unit. It is assumed that each production-decision unit has three input–output vectors of inputs, desired outputs, and nondesired outputs, which are represented by three matrices: $X = [x_1, \ldots, x_n] \in \mathrm{R}^{m \times n}, Y^g = [y_1^g, \ldots, y_n^g] \in R^{s_1 \times n}, Y^b = [y_1^b, \ldots, y_n^b] \in R^{s_2 \times n}; X > 0, Y^g > 0, Y^b > 0,$

The possible set of production is represented as

$$P = \left\{ (x, y^g, y^b) \middle| x \geq X\lambda, y^g \geq Y^g\lambda, \lambda \geq 0 \right\} \tag{1}$$

The set of production possibilities is denoted as

$$\rho^* = \min \frac{1 - \frac{1}{m} \sum_{i=1}^{m} \frac{S_i^-}{x_{i0}}}{1 + \frac{1}{S_1 + S_2} \left( \sum_{r=1}^{S_1} \frac{S_r^g}{y_{r0}^g} + \sum_{r=1}^{S_2} \frac{S_r^b}{y_{r0}^b} \right)} \tag{2}$$

$s.t.$
$x_0 = X\lambda + s^-; y_c^g = Y^g\lambda - s^g; Y_0^b = Y^b\lambda + s^b;$
$s^- \geq 0, s^g \geq 0, s^b \geq 0, \lambda \geq 0$

among them, $s^-$, $s^g$, and $s^b$ represent the values of the input overuse, expected output underproduction, and excessive pollution emissions, respectively. The objective $s^b$ function $\rho^*$ satisfies $0 \leq \rho^* \leq 1$ and is strictly decreasing with respect to $s^-$, $s^g$. If the decision-making unit is effective, it is $\rho^* = 1$, and $s^{-*} = 0, s^{b*} = 0, s^{g*} = 0$ is also satisfied.

The expression of the forestry green overall factor productivity from t to t + 1 may be generated by utilizing the ML index to calculate the green overall factor productivity based on the findings of the SBM model discussed above:

$$ML_t^{t+1} = \left\{ \frac{[1 + \overrightarrow{D_{0t}}(x_t, y_t^g, y_t^b; g_t)]}{[1 + \overrightarrow{D_t}(x_{t+1}, y_{t+1}^g, y_{t+1}^b; g_{t+1})]} \times \frac{[1 + \overrightarrow{D_{t+1}}(x_t, y_t^g, y_t^b; g_t)]}{[1 + \overrightarrow{D_{t+1}}(x_{t+1}, y_{t+1}^g, y_{t+1}^b; g_{t+1})]} \right\}^{\frac{1}{2}} \tag{3}$$

According to the relevant literature, this paper assumes that the *ML* indices of the provinces and cities in the base period are all 1, and the *ML* indices of the subsequent

years are obtained by cumulative multiplication with the indices of the corresponding years [30–32].

### 3.2.2. Fixed-Effect Panel Model

Compared to general linear regression models, the panel data model has the advantage of considering both the commonality of cross-sectional data and the individual-specific effects of the cross-sectional factors in the model. The t green overall factor productivity of forestry in the i province and city in the year is expressed as $GTFP_{i,t}$. The command control environmental regulation is expressed as $CE_{i,t}$, the market-incentive-based environmental regulation is expressed as $IE_{i,t}$, and the public participatory environmental regulation is expressed as $PE_{i,t}$:

$$\ln GTFP_{i,t} = \alpha + \beta_1 \ln CE_{i,t} + \beta_2 \ln IE_{i,t} + \beta_3 \ln PE_{i,t} + \beta_4 X_{i,t} + \mu_i + \varepsilon_{i,t} \tag{4}$$

where $X_{i,t}$ is the covariate matrix, $\mu_i$ represents the fixed effect of the individual, and $\varepsilon_{i,t}$ is the perturbation term.

### 3.2.3. Panel Smooth Transformation Model (PSTR)

The PSTR model is an extension of the Smoothed Transformation Model (STR) and the Panel Threshold Model (PTR), which enable the model coefficients to vary continuously with the change in the transformation variables by introducing a continuous transformation function to replace the discrete transformation function in the PTR. At the same time, the PSTR model can also better reflect the heterogeneity characteristics of the data cross-section and time. In this paper, the PSTR model is used for empirical research, and the model is constructed as follows:

$$GTFP_{i,t} = \mu_i + \beta_{0,i}ER_{i,t} + \beta_{1,i}X_{i,t} + \sum_{j=1}^{r} [(\beta_{0,i}^{j}ER_{i,t} + \beta_{2,i}X_{i,t}) \times g(ER_{i,t};r_j;c^j)] + \varepsilon_{i,t} \tag{5}$$

Among them, the $ER_{i,t}$ represents various types of environmental regulation and the $X_{i,t}$ is a set of control variables. The $g(ER_{i,t};r_j;c^j)$ is a smooth transition function, bounded and continuous, with a value between 0 and 1. When 0 or 1 is the $g(ER_{i,t};r_j;c^j)$ selected, it represents that the model is in the low or high system states. When its value changes smoothly between 0 and 1, the model changes between the low and high systems accordingly. The smoothing parameter $r_j$ and position parameter c can intuitively represent the $g(ER_{i,t};r_j;c^j)$ speed and threshold value of the conversion function between low and high systems. The $\mu_i$ represents the fixed effect of an individual, and the $\varepsilon_{i,t}$ is a perturbation term. This study selects different types of environmental regulations as the transformation variables.

Before establishing the PSTR model, a nonlinear effect test should first be conducted, namely a linear test and a residual nonlinear test. The linear test hypotheses are H0: r = 0 and H1: r = 1. If the original hypothesis is rejected, it indicates a nonlinear relationship between the environmental regulation and forestry green overall factor productivity, and there is at least one location parameter c; the residual nonlinear test hypotheses are H0: r = 1 and H1: r = 2, where r is the number of conversion functions. If the original hypothesis cannot be rejected, it indicates that the PSTR model only has one positional parameter c. According to the approach proposed by Gonzalez et al., the validity of linear and nonlinear statistics was assessed by using the Lagrange multiplier, while the likelihood ratio was employed to test the accuracy of LRT statistics. The specific formulation is presented below:

$$LM = \frac{TN(SSR_0 - SSR_1)}{SSR_0} \sim \chi^2(k) \tag{6}$$

$$LMF = \frac{(SSR_0 - SSR_1)/k}{SSR_0/(TN - N - k)} \sim F(k, TN - N - k) \tag{7}$$

$$LRT = -2\log\frac{SSR_1}{SSR_0} \sim \chi^2(k) \qquad (8)$$

where $k$ is the number of explanatory variables, $SSR_0$ is the sum of squares of residuals of the H0 hypothesis, and $SSR_1$ is the sum of squares of residuals of the alternative H1 hypothesis [33].

### 3.3. Variable Description

The explanatory variable is the forestry green overall factor productivity (GTFP), including the inputs, desired outputs, and nondesired outputs. To ensure the authenticity and credibility of the forestry green overall factor productivity measurements, the economic, ecological, and green efficiency are comprehensively considered while constructing the forestry input–output system shown in Table 1 below.

**Table 1.** Input–output system of green overall factor productivity in forestry.

| Type | | Variable Name | Data Description |
|---|---|---|---|
| Interpreted variable | Overall factor productivity of forestry | | |
| | | Investment | |
| | | Land input | Forestry land area |
| | | Energy input | Overall area energy consumption $\times$ gross regional forest product/gross regional product (GDP) |
| | | Labor input | Number of employees in the regional forestry industry at the end of the year |
| | | Capital investment | Forestry fixed assets investment stock |
| | | Expected output | |
| | | Economic performance | Gross regional forestry production |
| | | Ecological benefit | Area of afforestation in the current year |
| | | Unexpected output | |
| | | Regional forestry industry wastewater discharge | Regional industrial wastewater discharge$\times$ regional forestry secondary industry output value/regional industrial output value |
| | | Regional forestry industry $SO_2$ emissions | Regional industrial $SO_2$ emissions $\times$ regional forestry secondary industry output value/regional industrial output value |
| | | Regional forestry solid waste production | Regional generation of industrial solid waste $\times$ regional forestry secondary production value/gross regional industrial product |

The explanatory variable is the environmental regulation. The environmental regulation is a combination of various policies formulated or implemented by organizations or individuals to protect the environment and conserve resources, which is classified in this paper into command-and-control, market-incentive, and public-participation environmental regulations according to the difference in their management subjects.

Command-and-control environmental regulation (CE) refers to environmental regulations or pollution emission standards formulated or issued by the government and related

departments. Previous studies have mostly used the single-indicator substitution method, and some scholars have used the composite index method and the assignment-scoring method to measure the command-and-control type of environmental regulation. Considering that the single-indicator substitution method is not comprehensive enough and the assignment scoring method is subjective, this paper adopts the comprehensive index method to construct the environmental regulation index system, specifically five indicators, namely the industrial fume (dust) removal rate, the industrial sulfur dioxide removal rate, comprehensive utilization rate of industrial solid waste, centralized treatment rate of wastewater treatment plants, and harmless treatment rate of domestic garbage. It adopts the entropy method to calculate the command-and-control environmental regulation index. The stronger the government's control over the environment, the higher the regional index and vice versa [34].

Market-incentive-based environmental regulation (IE) requires the government to first formulate relevant policies through market regulation that guide enterprises in pursuing their interests to maximize while simultaneously completing the environmental protection task. This paper expresses it by the ratio of the sewage fee revenue to the gross regional product [35].

Public participatory environmental regulation (PE) mainly relies on the people's spontaneous response to environmental problems to the higher authorities, which can enable the government to hear the voice of the public and formulate more efficient environmental norms. In this paper, the number of suggestions and proposals on environmental protection from the National People's Congress and the Chinese People's Political Consultative Conference are used to characterize the PE [36].

The control variables are used as the interprovincial urbanization level (UL), GDP per capita (GDP), years of schooling per capita (PYE), and degree of openness to the outside world (OU), which are characterized by using the overall number of imports and exports ratio nominal GDP [37].

The study period in this paper was between 2006 and 2021. The samples are 11 provinces and municipalities along China's Yangtze River Economic Belt. The raw data of all the variables were obtained from the China Statistical Yearbook, China Environmental Statistical Yearbook, and China Forestry Statistical Yearbook. Due to the lack of environmental regulation index data in recent years, this paper adopts a linear interpolation method to supplement the missing data according to the relevant literature [38,39].

## 4. Results

### 4.1. Panel Data Model Regression Results

The benchmark regression results are presented in Table 2 below. This work was based on the Hausmann test employing a fixed-effects panel data model. Models 1 through 3 investigate how each environmental regulation affects the Yangtze River economy's overall green factor productivity of forestry, Model 4 investigates how three different environmental regulations simultaneously affect this productivity, and Model 5 is based on Model 4 by incorporating control variables. The regression results show that the command-and-control and public-participation environmental regulations are significantly positive, while the market-incentive environmental regulations have a dampening effect on the forestry green overall factor productivity, the regression coefficients of the variables are unchanged when the three types of environmental regulations are simultaneously estimated, and the regression coefficients of the variables are still unchanged after adding the control variables [40].

**Table 2.** Panel data model regression results.

| Variable | 1 | 2 | 3 | 4 | 5 |
|---|---|---|---|---|---|
| | GTFP | | | | |
| CE | 0.576 *** | | | 0.244 ** | 0.409 ** |
| | (5.022) | | | (2.408) | (2.181) |
| IE | | −0.0425 *** | | −0.020 | −0.018 |
| | | (−3.3895) | | (−1.668) | (−1.364) |
| PE | | | 0.109 *** | 0.027 | 0.012 |
| | | | (4.439) | (0.760) | (0.321) |
| Control variable | × | × | × | × | ✓ |
| R-squared | 0.689 | 0.751 | 0.755 | 0.780 | 0.776 |

Note: ** and *** indicate significance at 5% and 1%, respectively, with *t* values in parentheses. Taking Model 5 as an example, the Hausman test *p* value was 0.000, less than 0.05, so the fixed-effect panel model was ultimately selected.

*4.2. Spatiotemporal Heterogeneity Test*

The panel data model regression results preliminarily verify the research hypothesis of this paper. However, there is a need to further verify whether there are differences in the regression results across time or regions. Based on this, the following Table 3 shows the results of the temporal heterogeneity test. The regression results demonstrate that the command-and-control and market-incentive environmental regulation regression coefficients are both significantly positive in the lower reaches of the Yangtze River and that the command-and-control environmental regulation regression coefficients are significantly positive in the middle reaches of the Yangtze River. The Yangtze River's upper reaches require more improved environmental regulations than the lower reaches, and the middle and upper reaches of the river require more improved environmental regulation than the lower reaches due to the concentration of highly polluting industries [41]. This is evident from the spatial heterogeneity test.

**Table 3.** Test results for spatiotemporal heterogeneity.

| Variable | GTFP | | | | |
|---|---|---|---|---|---|
| | Regional Heterogeneity | | | Temporal Heterogeneity | |
| | Upper Yangtze River | The Middle Reaches of the Yangtze River | Lower Yangtze River | Before 2014 | After 2014 |
| CE | 0.276 | 1.591 ** | 0.939 ** | 0.069 | 0.178 * |
| | (0.913) | (3.058) | (2.330) | (0.511) | (1.942) |
| IE | −0.007 | 0.082 | 0.138 ** | −0.010 | −0.013 * |
| | (−0.521) | (0.630) | (2.503) | (−0.152) | (−1.81) |
| PE | −0.103 | 0.077 | −0.151 ** | −0.008 | 0.027 * |
| | (−1.340) | (1.071) | (−2.022) | (−0.163) | (−0.806) |
| Control variable | ✓ | ✓ | ✓ | ✓ | ✓ |
| R-squared | 0.798 | 0.774 | 0.572 | 0.001 | 0.091 |

Note: * and ** indicate significant at the levels of 10% and 5%, respectively, with *t* values in parentheses.

In 2014, the central government's work report elevated the development of China's Yangtze River Economic Belt to a national strategy; based on this, this study takes 2014 as the boundary to compare the temporal heterogeneity of the level of environmental regulation impacts before and after the implementation of the strategy. The results are shown in Table 3 below. Compared with the period before the development strategy was proposed, the promotion effect was strengthened, and the influence of environmental legislation on the green overall factor production in forestry was greater. As a result, hypothesis 1 was accepted.

### 4.3. Panel Smoothing Transformation Regression (PSTR) Model Estimation Results

#### 4.3.1. Test Results of Nonlinear Effects

Using various types of environmental regulations as conversion variables and core explanatory variables, we apply the PSTR model to empirically test the nonlinear relationship between environmental regulations and the green overall elements of forestry in the Yangtze River Economic Belt of China. However, we first need to conduct a nonlinear effect test, namely a linear test and a residual nonlinear test. The specific formulation is presented below:

It is evident from the test results in the accompanying table that with three types of environmental regulation as the transformed variables, all the results of the nonlinear tests significantly reject the original hypothesis at the 1% level, indicating that the panel smoothed transformed regression model has at least one location parameter, and at the same time, indicating that this model is applicable for this study. It is evident from the residual nonlinear test findings in Table 4 above that the original hypothesis that the model has one location parameter cannot be rejected. Therefore, this study applies the panel smoothed transformed regression model with two regimes to measure the nonlinear relationship between environmental regulations and the production of the green overall factor in forestry [42].

**Table 4.** Nonlinearity test results.

| Transformation Variable | Nonlinear Test (H0: r = 0, H1: r = 1) | | | Residual Nonlinear Test (H0: r = 1, H1: r = 2) | | |
|---|---|---|---|---|---|---|
| | LM | LMF | LRT | LM | LMF | LRT |
| CE | 46.879 *** | 11.618 *** | 54.513 *** | 1.831 | 0.315 | 1.840 |
| | (0.000) | (0.000) | (0.000) | (0.872) | (0.903) | (0.871) |
| IE | 32.918 *** | 7.362 *** | 36.443 *** | 0.867 | 0.900 | 0.866 |
| | (0.000) | (0.000) | (0.000) | (1.865) | (0.321) | (1.875) |
| PE | 39.879 *** | 9.375 *** | 45.221 *** | 13.704 | 2.533 | 14.267 |
| | (0.000) | (0.000) | (0.000) | (0.018) | (0.031) | (0.014) |

Note: *** indicate significance at 1%, with $p$ values in parentheses.

#### 4.3.2. PSTR Model Estimation Results

This study used the green overall factor productivity of forestry as the explanatory variable. Three types of environmental regulations were used as the transforming and core explanatory variables. These variables were measured by using MATLAB R2018a and the nonlinear least squares method. The estimation results of Equation (5) were as follows (Table 5).

**Table 5.** PSTR model estimation results.

| Variable | CE | | IE | | PE | |
|---|---|---|---|---|---|---|
| | Low System | High System | Low System | High System | Low System | High System |
| ER. | −0.694 * | 0.629 * | 0.018 | 0.829 * | 0.214 * | −0.338 *** |
| | (1.784) | (1.741) | (0.478) | (1.746) | (1.506) | (−3.203) |
| UL. | −2.412 * | 0.851 *** | −5.756 *** | 3.665 *** | −0.034 | −2.375 *** |
| | (2.352) | (3.391) | (−4.898) | (3.755) | (−0.043) | (−3.543) |
| GDP | 3.380 | 2.221 | −1.524 *** | 0.316 *** | −0.400 * | −0.111 * |
| | (0.687) | (0.870) | (−4.020) | (3.756) | (−1.183) | (0.982) |
| DO | −0.365 | 0.464 | 0.943 *** | −0.323 ** | 0.214 * | 0.212 |
| | (0.416) | (0.567) | (6.601) | (2.460) | (1.367) | (−0.011) |
| PYE | −4.167 | 4.866 * | 7.416 *** | 3.323 ** | 0.940 * | 0.531 |
| | (−0.593) | (1.826) | (4.347) | (−2.564) | (0.726) | (−0.285) |
| r | 3.011 | | 0.911 | | 45.800 | |
| c | 3.778 | | 6.064 | | 6.240 | |

Note: *, **, and *** indicate significance at 10%, 5%, and 1%, respectively, with $t$ values in parentheses.

When command-and-control environmental regulation is used as the conversion variable, a threshold value of 3.778 corresponds to a command-and-control environmental regulation index of 43.73%. Before and after the threshold value, the influence mechanism of the command-and-control environmental regulation on the green overall factor productivity in forestry changes from inhibition to promotion. That is, command-and-control environmental regulation will have a promoting influence on the forestry overall factor productivity only when it reaches a certain intensity. When environmental regulation is low, businesses will divert resources from other projects to treat pollution and cut emissions in the short term, lowering productivity. On the other hand, when the level of environmental regulation is higher, businesses will invest directly in green innovation for the long term, allowing them to improve productivity while cutting emissions of pollutants [43].

Taking market-incentive-based environmental regulation as a conversion variable, a threshold value of 6.046 exists, which indicates that the ratio of the sewage fee revenue to the gross regional product is 0.042%. Above the threshold value, the positive contribution of the market-incentive-based environmental regulation to the green overall factor productivity in forestry increases. Some businesses increase R&D expenditure for green innovation technology enhancement in response to the intensity of environmental legislation, or they invest in clean sectors to boost their competitiveness [44].

Taking the public participation type of environmental regulation as a transforming variable, the existence of the threshold value of 6.240 represents the number of suggestions and proposals on environmental protection from the National People's Congress (NPC) and the Chinese People's Political Consultative Conference (CPPCC), which is roughly 513. Before and after the threshold value, the mechanism of public participatory environmental regulation's influence on the green overall factor productivity in forestry is changed from promotion to inhibition. Therefore, appropriate public participatory environmental regulation can restrain the behavior of enterprises without causing excessive pressure on enterprises, which can reduce pollutant emissions while eliminating the backward production capacity. Productivity is maintained at a high level [45]. Based on these findings, Hypothesis 2 is accepted. The transformation function is shown in Figure 2 below.

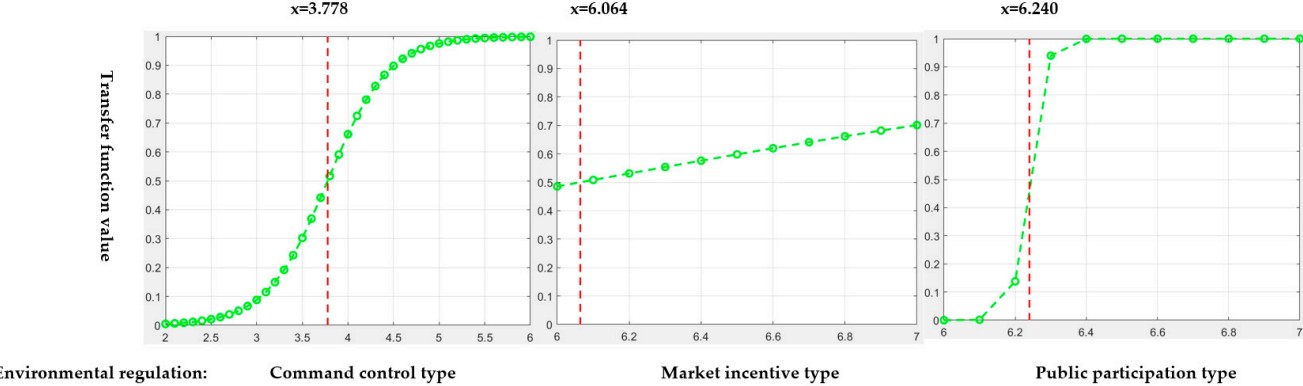

**Figure 2.** Transfer function image.

## 5. Discussion

Through the collation of related research findings domestically and abroad, research on the implications of the mechanism of environmental regulation on the overall green factor productivity is found to involve the manufacturing, agriculture, services, and mining industries. The adopted research methodologies include the threshold effect, long dynamic panel model, and spatial econometric model. The research perspectives are guided and classified according to three theories: "innovation compensation theory", "compliance theory", and "green overall factor productivity theory" [46–49].

"Compensation for innovation theory".

The "innovation compensation theory" refers to the idea that timely, suitable environmental regulations can help boost the productivity of green overall factors. Porter put forward the "innovation compensation effect" from the dynamic perspective [50]. At the same time, it is noted that both direct and indirect effects make up the majority of the positive promotion impact of environmental regulation on the green overall factor output [51]. The term "direct effect" refers to the possibility that increased environmental legislation may immediately result in lower energy consumption and corporate pollution emissions. The "indirect effect" refers to the promotion of the green overall factor productivity through incentivizing green technological innovation in products and promoting the green overall factor productivity through technological innovation [52]. Porter's hypothesis suggests that appropriate environmental regulation can stimulate technological innovation and motivate enterprises to carry out more innovative initiatives and thus offset the costs of environmental protection and enhance the profitability of enterprises [53].

"Compliance cost argument"

The "cost of compliance" theory refers to the implementation of strict environmental regulations by the relevant departments, increasing sewage charges, which in turn restricts the technological innovation initiatives of enterprises, thus inhibiting the enhancement of the green overall factor productivity. Unlike "Porter's hypothesis", this viewpoint suggests that environmental regulation does not produce a compensatory innovation effect to increase the green overall factor productivity but rather increases the pressure on the cost of pollution control [54].

"Uncertainty theory"

With the continuous deepening of related research, many scholars have shown that there is an uncertain relationship between environmental regulation and the overall green factor productivity [55]. Factors such as the type of environmental regulation, the intensity of regulation, and the length of the implementation time will affect the mechanism of its impact on the green overall factor productivity [56].

As expected, the findings of the present study suggest that the mechanism of environmental regulation on the green overall factor productivity in forestry varies across different types, regulatory intensities, periods, and regions.

## 6. Conclusions

### 6.1. Main Conclusion

In this study, relevant data from 11 provinces along the Yangtze River Economic Belt in China from 2006 to 2021 are selected to study the linear and nonlinear relationships and spatiotemporal heterogeneity of command-and-control, market-incentive, and public-participation types of environmental regulations with the green overall factor productivity in forestry. This study found the following:

When compared with market-incentive-based environmental regulations, command-and-control and public participation-based environmental regulations have stronger constraints, which can significantly enhance the overall factor productivity of forestry in the Yangtze River Economic Zone. Environmental regulations make enterprises have to invest in green technological innovation so that the green overall factor productivity of forestry in the industry can be improved in the long run.

From a regional perspective, the implementation effect of environmental regulations in the lower reaches of the Yangtze River is better than that in the middle reaches. The effect of environmental regulations in the middle reaches of the Yangtze River is better than that in the upper reaches of the Yangtze River, showing an increasing trend in the order of the upper, middle, and lower reaches. This may be due to the favorable geographical location, developed economic level, and high technology level of the lower reaches of the Yangtze River.

From the perspective of time, the development of the Yangtze River Economic Belt was upgraded to a national strategy in 2014, after which command-and-control and public-participation environmental regulations promoted the green overall factor productivity.

Further research shows that the promotion effect of environmental regulation on the green overall factor productivity of forestry in the Yangtze River Economic Belt has nonlinear characteristics:

There is a threshold value of 43.73% for the influence mechanism of command-and-control environmental regulations on the forestry green overall factor productivity. The influence mechanism turns from negative to positive before and after the threshold value.

There is a threshold value of 0.042% for the influence mechanism of market-incentive-type environmental regulations on the green overall factor productivity in forestry. After exceeding the threshold value, its positive promotion effect is enhanced. At the same time, the urbanization level and GDP per capita have a similar shift.

There is a threshold value of 513 for the influence mechanism of public-participation-type environmental regulation on the green overall factor productivity in forestry. After exceeding the threshold value, its influence mechanism turns positive to negative, i.e., the intensity of public-participation-type environmental regulations should not be too high.

*6.2. Policy Recommendations*

Based on the above conclusions, the following policies are proposed for reference to improve the green overall factor productivity of forestry in the Yangtze River Economic Belt more efficiently. This could make it become the main battlefield for China's ecological prioritization and green development and lead to the high-quality development of the economy.

Command-and-control and public-participation types of environmental regulation have significantly positive effects on the green overall factor productivity in forestry in the Yangtze River Economic Belt. The government should focus on coordinating the use of different types of environmental regulations, continue to play the role of command-and-control environmental regulations in promoting the green overall factor productivity of forestry in China's Yangtze River Economic Belt, and improve the relevant laws and regulations to encourage producers to take the initiative to perform source control rather than just end-of-pipe penalties. While improving the operability of relevant laws and regulations, the government should use the public participation type of environmental regulation to promote technological innovation.

Based on the characteristics of the Yangtze River's upper, middle, and lower reaches, different environmental regulations should be implemented in different areas according to the different resources and actual conditions of the areas. For the ecologically sound downstream region with a high level of economic development, it is necessary to strengthen the technological innovation and reinforce the ecological barrier of China's Yangtze River Economic Belt. This can be achieved while simultaneously maintaining the environmental advantages and establishing sound environmental protection laws and regulations due to different resource endowments. The upper and middle reaches of the Yangtze River are relatively backward in terms of economic development, and they should utilize a variety of environmental regulatory means in a coordinated manner to realize high-quality economic development while coordinating to promote the protection of the ecological environment.

Surpassing the threshold value results in a similar shift in the influence mechanism of the urbanization level and GDP per capita on the green overall factor productivity in forestry in the Yangtze River Economic Belt against the market-incentive-based environmental regulation. Therefore, the exploration of the appropriate market-incentive-based environmental regulation intensity should be combined with the enhancement of the urbanization level and GDP per capita to enhance the green overall factor productivity in forestry from the perspectives of the whole society, the whole factor, and the entire industry.

*6.3. Research Deficiencies and Prospects*

Drawing on the basis of the results of previous studies, the corresponding preliminary conclusions are obtained through empirical analysis. Shortcomings can be diverse and can be attributed to the author's theoretical level, data availability, and other constraints. The present study has certain shortcomings, which necessitates more in-depth refinements of the study [57–60].

The selection of environmental regulation indicators needs further improvement. At present, the level of environmental regulation in China is constantly developing. The actual operation will often be implemented in conjunction with a variety of policies, and the differences in environmental regulation in various provinces often lead to more difficulties in quantifying the level of environmental regulation. The present study selected a representative index. However, the resulting index cannot fully reflect the level of environmental regulation.

Given the wide range of theories, perspectives, and views from other studies, the extent of the work conducted in the present study may have been relatively less comprehensive.

**Author Contributions:** D.L. and G.T. helped conceptualize the study design; Y.L. performed the statistical analyses and participated in writing, reviewing, and editing; and R.K.M. made equal contributions and provided his intellectual insights. All authors have read and agreed to the published version of the manuscript.

**Funding:** This research was funded by the National Social Science Foundation Project of China, grant number 21BGJ066.

**Data Availability Statement:** Publicly available datasets were analyzed in this study. These data can be found here: https://data-cnki-net-s.webvpn.nefu.edu.cn/yearBook/single?id=N2021060073, https://data-cnki-net-s.webvpn.nefu.edu.cn/yearBook/single?id=N2022030234, and https://www.epsnet.com.cn/index.html#/Index, (accessed on 1 December 2022).

**Acknowledgments:** The authors express their gratitude to everyone who helped make this research a success.

**Conflicts of Interest:** The authors declare no conflict of interest.

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
