# Peer review of "The Impact of Environmental Regulation on the Green Overall Factor Productivity of Forestry in the Yangtze River Economic Belt"

_forests, doi:10.3390/f14102004_

Round 1

Reviewer 1 Report

The paper is a research article that studies the influence effect of environmental regulation on green total factor productivity of forestry in the Yangtze River Economic Belt.

It is a well written paper with proper structure and adequate literature review.

The research method is well described. 

The results are robust and the conclusions are based on the results.

The literature review section could be enriched with more references.

The numbering of the sub-sections needs to be correcter. E.g. Line 198 3.2 instead of 2.2 etc.

The References should be written according to the Instructions for Authors of the journal Forests.

Author Response

Dear reviewer:

We feel great thanks for your valuable feedback that we have used to improve the quality of our manuscript. We have studied comments carefully and have made correction which we hope meet with approval. Revised portion are marked in yellow in the paper. The main corrections in the paper and the responds to the review’s comments are as flowing:

Valuable comments:

  1. The literature review section could be enriched with more references.
  2. The numbering of the sub-sections needs to be correcter. E.g. Line 198 3.2 instead of 2.2 etc.
  3. The References should be written according to the Instructions for Authors of the journal Forests.

Response to comment:

  1. Based on your valuable suggestions, we carefully checked the references and added more relevant literature at the beginning of line 155. The reviewers have asked question about the same content, which I have highlighted in grey in the article.
  2. Sorry for our carelessness. In the resubmitted manuscript, we have made revisions on 285 lines. Thanks for the correction.
  3. Thank you very much for your suggestions, we have revised the references to the format of the journal Forests.

Once again, thank you very much for your comments and suggests.

Thank you and beat regards.

Sincerely Liyang

Reviewer 2 Report

The article leaves the impression of a pretty good piece of work that is so far rather mediocre in its description.

The topic of assessing the effects of environmental regulation on the economic efficiency of forestry is quite popular in modern literature. For these purposes, it is quite common to apply methods of operations research, including methods of data envelopment analysis, which are applied in the peer-reviewed article. 

However, the article must be substantially revised.

1. The article contains many errors in the design and wording of the text (unnecessary and inappropriate word combination at the end of key words, the list of sources is arranged against the rules of the journal, unnecessary words in the description of funding, etc.).

2. Statistical conclusions should be carefully checked for correctness. For example, statements like "At this point, hypothesis 1 has been verified" (line 332) are incorrect. It is correct to speak about rejection or impossibility to reject a hypothesis with a certain level of statistical significance. Such examples should be checked throughout the manuscript.

3. The literature review is too technical and does not contain a proper analysis. Specific results like "such-and-such author concluded this and that" can be aggregated into tables. They should be summarized in the text.

Native-speaker editing is required. 

Author Response

Dear reviewer:

We feel great thanks for your valuable feedback that we have used to improve the quality of our manuscript. We have studied comments carefully and have made correction which we hope meet with approval. Revised portion are marked in red in the paper. The main corrections in the paper and the responds to the review’s comments are as flowing:

Valuable comments:

  1. The article contains many errors in the design and wording of the text (unnecessary and inappropriate word combination at the end of key words, the list of sources is arranged against the rules of the journal, unnecessary words in the description of funding, etc.).
  2. Statistical conclusions should be carefully checked for correctness. For example, statements like "At this point, hypothesis 1 has been verified" (line 332) are incorrect. It is correct to speak about rejection or impossibility to reject a hypothesis with a certain level of statistical significance. Such examples should be checked throughout the manuscript.
  3. The literature review is too technical and does not contain a proper analysis. Specific results like "such-and-such author concluded this and that" can be aggregated into tables. They should be summarized in the text.

Response to comment:

  1. Thanks for your suggestion. We do invited a friend of us who is a native English speaker to help polish our article. And we hope the revised manuscript could be acceptable for you.
  2. Thank you for your valuable comments, based on empirical assumptions, we have revised lines 434 and 508 of the newly submitted manuscript accordingly.
  3. Based on your valuable suggestions, we carefully checked the references and added more relevant literature at the beginning of line 155. The reviewers have asked question about the same content, which I have highlighted in grey in the article.

Once again, thank you very much for your comments and suggests.

Thank you and beat regards.

Sincerely Liyang

Reviewer 3 Report

First of all, I appreciate the opportunity to review the paper Research on the influence effect of environmental regulation on green total factor productivity of forestry in the Yangtze River Economic Belt.  The paper deals with a very interesting problem.

Suggestions are below:

·        The abstract is not well written. The most important results/findings must be emphasized.

·        Why the observation period does not include 2021 and 2022?

·   Keywords are not well defined. For example, this is not a keyword: “green total factor productivity in forestry”.

·        The last paragraph in the introduction section is a short structure of the paper (several sentences for each section).

·        It is necessary to understand the purpose and aim of the paper as well as its "position" in relation to previous research (also gap analysis).

·        The separate section Practical and theoretical implications (or Discussion) is missing. This confirms the lack of scientific and practical contribution.

·        Conclusion section is not on a satisfactory level. The conclusion in scientific papers is very important.

o   Limitations of your research must be emphasized

o   Future research directions are missing.

·        Scientific contributions are questionable.

Suggested references:

·        Andrejić M., Kilibarda, M. Pajić, V., (2021). Measuring efficiency change in time applying Malmquist productivity index: a case of distribution centres in Serbia. FACTA UNIVERSITATIS, Series Mechanical Engineering, 19 (3), 499-514.

·        Qin, W.; Qi, X. Evaluation of Green Logistics Efficiency in Northwest China. Sustainability 2022, 14, 6848. https://doi.org/10.3390/su14116848

·        Zhao, W.; Qiu, Y.; Lu, W.; Yuan, P. Input–Output Efficiency of Chinese Power Generation Enterprises and Its Improvement Direction-Based on Three-Stage DEA Model. Sustainability 2022, 14, 7421. https://doi.org/10.3390/su14127421 

.

Author Response

Dear reviewer:

We feel great thanks for your valuable feedback that we have used to improve the quality of our manuscript. We have studied comments carefully and have made correction which we hope meet with approval. Revised portion are marked in green in the paper. The main corrections in the paper and the responds to the review’s comments are as flowing:

Valuable comments:

  1. The abstract is not well written. The most important results/findings must be emphasized.
  2. Why the observation period does not include 2021 and 2022?
  3. Keywords are not well defined. For example, this is not a keyword: “green total factor productivity in forestry”.
  4. The last paragraph in the introduction section is a short structure of the paper (several sentences for each section).
  5. It is necessary to understand the purpose and aim of the paper as well as its "position" in relation to previous research (also gap analysis).
  6. The separate section Practical and theoretical implications (or Discussion) is missing. This confirms the lack of scientific and practical contribution.
  7. Conclusion section is not on a satisfactory level. The conclusion in scientific papers is very important.

Limitations of your research must be emphasized

Future research directions are missing.

Response to comment:

  1. Thank you for your valuable suggestions, we have rewritten the abstract section in the new submission.
  2. We very much appreciate your suggestions and have updated the data in the new submission based on the latest Forestry Statistical Yearbook.
  3. Thank you for your valuable comments, we have revised the keywords accordingly in the new submission.
  4. Thank you for your valuable advice, we have done our best to explain this in line 227 of the new submission.
  5. We think this is a good suggestion. Following the reviewers' recommendations, we added research purpose to line 125 of the newly submitted manuscript and summarized the position of this paper in lines 260 and 283 based on existing research.
  6. We are very sympathetic to your suggestion, and the 87 lines in the newly submitted manuscript supplement the theoretical and practical significance of this study.
  7. Thank you very much for your suggestions, we did our best to add relevant content to the 663 lines of the new manuscript submission.

Once again, thank you very much for your comments and suggests.

Thank you and beat regards.

Sincerely Liyang

Reviewer 4 Report

The Authors of the Article analyzed the impact of environmental regulations on green total factor productivity in forestry in the Yangtze River Economic Belt of China. Based on the relevant data of 11 provinces along the Yangtze River Economic Belt in P.R. China from 2006 to 2020, this paper uses the ultra-efficient undesired output SBM-ML index model to measure the green total factor productivity of forestry in the Yangtze River Economic Belt, and uses ordinary panel regression and panel smooth transformation models to study the influence effect of environmental regulation on forestry green total factor productivity.

The topics of the article are current and in line with actual research trends.

As for the structure and layout of the article, I miss the discussion of the results. Please add such a chapter before the Conclusions chapter and include a broader analysis of the results obtained, comparing them with the results obtained by other researchers, in other areas, using different methodology.

Comments on the content of the article:

1 In the Literature Review chapter, the authors refer only to the findings of Chinese researchers. I miss the presentation of studies from other countries here. As it stands, the article is too local in nature.

2. I miss the description of the subject of the research - please add a subsection describing the analyzed area in terms of information related to the research conducted, it would also be good to add a figure showing the location of the area.

3. I don't quite understand the location of points (1) and (2) (lines 181-195). The first one is like a conclusion that is not clear where it comes from. The second is like a justification for the formulation of hypothesis 2. With the layout as it is now, these points do not quite fit here in this way. Please rewrite them.

4. Repeatedly in the article appears the phrase "environmental regulations". The authors describe it in a fairly general way in re Introduction, a little more detail in section 2.3 Description of the variable. However, I miss a more detailed description of this element here. What exactly consists of CE, IE and EP? What laws, regulations, standards, etc.? Please add this in the article.

5. line 347: please elaborate on the acronyms LM, LMF, and LRT. Nowhere in the article is this mentioned, and they may not be obvious to every reader....

6. Please link the presented conclusions to the research results in a more direct way (lines 427-463). Please clearly show which results allow you to formulate which conclusion. In the current version, the conclusions are not directly linked to the results of the presented research, and in fact each conclusion contains a reference to the literature. This fits more with a discussion of results rather than conclusions. The Conclusions chapter needs a major rewrite.

Minor editorial comments:

1. the numbering of subsections is confused - in Chapter 3 there are subsections: 2.2, 2.2.1, 2.2.2, 2.2.3, 2.3

2. the numbering of tables is confused - Table 2 appears twice, Table 3 is missing

3. please remove that Chinese sign in Table 4

I am not a language specialist, but according to me, the article needs language correction (e.g. lines 147-148 - this is not a correct sentence).

The article is interesting, but needs improvement.

Author Response

Dear reviewer:

We feel great thanks for your valuable feedback that we have used to improve the quality of our manuscript. We have studied comments carefully and have made correction which we hope meet with approval. Revised portion are marked in purple in the paper. The main corrections in the paper and the responds to the review’s comments are as flowing:

Response to comment:

  1. We very much appreciate your suggestion and add a discussion of the results to the 513 lines of the new submission.
  2. We very much agree with you and add relevant English literature reviews at the beginning of the 160 lines of the new manuscript. The reviewers have asked question about the same content, which I have highlighted in grey in the article.
  3. Thank you very much for your suggestions, we have added a description of the study area and corresponding images in lines 41-59 of the new submission.
  4. We sincerely appreciate the valuable comments and we have re-written this part according to the reviewer’s suggestion. (Lines 266-282 of the newly submitted manuscript)
  5. Thank you very much for your suggestion, we have added a description of environmental regulation in lines 359-388 of the new submission.
  6. We very much accept your suggestion and have added a description of the test method to lines 449-459 of the new submission.
  7. Thank you for your valuable advice, we did our best to rewrite the policy recommendation in line 609 of the new submission.

Response to editorial comment:

  1. Sorry for our carelessness, we have revised the numbering of chapters and tables and removed the table 4 space in the new submission.
  2. Thanks for your suggestion. We do invited a friend of us who is a native English speaker to help polish our article. And we hope the revised manuscript could be acceptable for you.

Once again, thank you very much for your comments and suggests.

Thank you and beat regards.

Sincerely Liyang

Round 2

Reviewer 2 Report

I still insist that the title must be shortened. Please omit the "Research on" beginning.

Another tour of thorough editing of text would also be beneficial

Another tour of thorough editing of text would also be beneficial

Author Response

Dear reviewer:

We feel great thanks for your valuable feedback that we have used to improve the quality of our manuscript. We have studied comments carefully and have made correction which we hope meet with approval. Revised portion are marked in yellow in the paper. The main corrections in the paper and the responds to the review’s comments are as flowing:

Comment1:

I still insist that the title must be shortened. Please omit the "Research on" beginning.

Response to comment1:

We strongly agree with your suggestion and have revised the title.

Comment2:

Another tour of thorough editing of text would also be beneficial.

Response to comment2:

Thank you for your suggestion. We did invite a native English speaker to help polish our article. We hope that the revised manuscript can be accepted by you.

Once again, thank you very much for your comments and suggests.

Thank you and beat regards.

Sincerely Liyang

Reviewer 3 Report

The paper should be accepted for publication.

.

Author Response

Dear reviewer:

      We feel great thanks for your valuable feedback that we have used to improve the quality of our manuscript. I We have seen that your Review Report Form indicated that the results presentation and references need to be improved. Therefore, according to the suggestions of other reviewers, we made modifications and highlighted in the paper. In addition, We did invite a native English speaker to help polish our article.

Thank you again.

Sincerely

Liyang

Reviewer 4 Report

The article has been improved significantly. My comments have been taken into account. I have still only minor comments :

1. formulas (6), (7), (8) from subsection 4.3.1 please relocate to the previous chapter along with the description. Only the results are to remain in Chapter 4. The description: on what basis and why they were obtained should be in Chapter 3.

2. in Table 4 further there is some Chinese character (first line). If it is necessary, and it is not some typo, please explain this mark in the text.

I recommend the article for publication.

Author Response

Dear reviewer:

We feel great thanks for your valuable feedback that we have used to improve the quality of our manuscript. We have studied comments carefully and have made correction which we hope meet with approval. Revised portion are marked in red in the paper. The main corrections in the paper and the responds to the review’s comments are as flowing:

Comment1:

Formulas (6), (7), (8) from subsection 4.3.1 please relocate to the previous chapter along with the description. Only the results are to remain in Chapter 4. The description: on what basis and why they were obtained should be in Chapter 3.

Response to comment1:

We fully agree with your suggestion and have moved the corresponding theoretical content to Chapter 3 (line 345).

Comment2:

In Table 4 further there is some Chinese character (first line). If it is necessary, and it is not some typo, please explain this mark in the text.

Response to comment2:

Thank you for your suggestion. We have changed the relevant characters in the first row (line 464) of Table 4 to English format, and the relevant definitions of the symbols are in line 350.

Once again, thank you very much for your comments and suggests.

Thank you and beat regards.

Sincerely Liyang
